# Markers of Schizophrenia—A Critical Narrative Update

**DOI:** 10.3390/jcm11143964

**Published:** 2022-07-07

**Authors:** Beata Galińska-Skok, Napoleon Waszkiewicz

**Affiliations:** Department of Psychiatry, Medical University of Białystok, 15-089 Białystok, Poland; begal@poczta.onet.pl

**Keywords:** biomarker, marker, psychosis, schizophrenia, neuroimaging, neurophysiological, biochemical, multimodal, peripheral, central

## Abstract

Schizophrenia is a long-term mental disease, associated with functional impairment. Therefore, it is important to make an accurate diagnosis and implement the proper treatment. Biomarkers may be a potential tool for these purposes. Regarding advances in biomarker studies in psychosis, the current symptom-based criteria seem to be no longer sufficient in clinical settings. This narrative review describes biomarkers of psychosis focusing on the biochemical (peripheral and central), neurophysiological, neuropsychological and neuroimaging findings as well as the multimodal approach related with them. Endophenotype markers (especially neuropsychological and occulomotor disturbances) can be currently used in a clinical settings, whereas neuroimaging glutamate/glutamine and D2/D3 receptor density changes, as well as immunological Th2 and PRL levels, seem to be potential biomarkers that need further accuracy tests. When searching for biochemical/immunological markers in the diagnosis of psychosis, the appropriate time of body fluid collection needs to be considered to minimize the influence of the stress axis on their concentrations. In schizophrenia diagnostics, a multimodal approach seems to be highly recommended.

## 1. Introduction

The aetiology of schizophrenia has not yet been well recognised. Its aetiology is multifactorial and clinical symptoms should fulfil its diagnostic criteria (ICD-10 or DSM IV). Although an organic brain process should be ruled out, schizophrenia is currently considered to be a brain disease based upon a neurodevelopmental process [1]. The young mean age of onset—usually the second decade—along with its long-term course make schizophrenia a debilitating disease [2]. Scientists have been searching for the biomarkers of psychiatric disorders, but establishing these is a very difficult task [3]. The pathophysiology of psychiatric disorders is multifactorial as they have polygenic aetiologies and each gene has a small effect, meaning that they can be modified by epigenetic mechanisms and that the environment modulates the course of the disease. It seems, therefore, that researchers may identify groups of biomarkers instead of a specific biomarker. Despite the growth in research papers about biomarkers, there has not been a similar increase in their clinical use [4]. The biomarkers can be described as diagnostic, prognostic or theranostic—prediction of the treatment response. In addition, biomarkers may be connected to each other peripherally and/or centrally; therefore, blood-related biomarkers are also a helpful tool to reveal some processes in the brain [5].

This narrative review presents the latest research on biochemical, neurophysiological, neuropsychological and neuroimaging markers of schizophrenia.

## 2. Materials and Methods

The literature search was carried out in PubMed, Scopus and Google Scholar by using the keywords of “psychosis”, “schizophrenia”, “marker”, “biomarker”, and different combinations of these keywords. Papers were then incorporated with the purpose of describing the widest possible selection of potential markers of schizophrenia.

## 3. Results and Discussion

### 3.1. Peripheral and Cerebrospinal Fluid Biomarkers

Antipsychotic treatment can cause neuroendocrine and metabolic disturbances in patients with schizophrenia. The studies of first-episode psychosis (FEP) allow the exclusion of the long-term impact of treatment and consequences of the chronic illness process. Therefore, the evaluation of the early markers among these patients seems to elucidate the pathophysiology of schizophrenia.

Antipsychotic naïve FEP is connected with increased fasting and oral-tolerance test-related glucose as well as fasting insulin levels and insulin resistance, raised triglycerides as well as reduced total cholesterol and LDL cholesterol [6,7]. In addition, prolactin (PRL) levels are elevated in antipsychotic naïve patients with schizophrenia [6,8]. As the early psychosis raises hypothalamic-pituitary-adrenal axis (HPA) with an increase in the number and size of corticotroph cells, it may induce stress-related lipid, glucose, insulin and hormonal dysregulation [6]. However, an exaggerated PRL response may also possibly be due to dysfunction in the dopaminergic transmission [6]. Hence further studies need to check PRL applicability as a marker of schizophrenia, separately in males and females.

The abnormalities in HPA axis function may be linked with the development of psychosis. The cortisol awakening response (CAR) is flattened in patients with schizophrenia and FEP, but not in persons who are only at risk of a mental disorder [9]. These findings may indicate CAR as a marker for transition risk. Only men at ultra-high risk (UHR) for psychosis demonstrated flat cortisol levels within two hours after awakening, while UHR women had a pattern of cortisol secretion similar to healthy subjects [10]. It also is known that the later onset of schizophrenia is associated with early puberty in girls, lower relapse of psychiatric symptoms during pregnancy, high relapse postpartum, and fluctuation of the symptoms across the menstrual cycle, as well as with the exacerbation of psychotic symptoms in women with schizophrenia during the menopausal transition [11]. However, the exact mechanism (genetic, metabolic, neuroprotective, etc.) by which sex hormones influence the onset, course and outcome of schizophrenia is still not adequately elucidated.

Inflammation-related factors can also be involved in psychosis [12]. First episode psychosis is correlated with increased levels of blood cytokine, involving IL-1β, sIL2R, IL-6, TNFα, TGFβ and elevated number of total lymphocyte [7]. Other studies have also suggested an altered concentration of other proteins as markers of schizophrenia, e.g., IL-12, CRP, S100 proteins and nerve growth factor (NGF) [13]. As IL-1, IL-6, and TNF can be reduced while antipsychotic treatment, and their decrease can be the result of the reduced activity of the stress axis, [14], it has been suggested that potential immunological markers can be found after the partial normalization of the immunological markers of acute stress. In accordance with the “Immunoseasonal” theory of mental disorders, described by Waszkiewicz [14], to maintain Th1–Th2 (Th1-IL-1β, IL-2, IL-6, TNF-α, IFN-, and Th2-IL-4, IL-5 and IL-10) immune homeostasis during a mood-related increased Th1 response in schizophrenia patients, the body increases the Th2 late response that dominates when the Th1 response declines post-seasonally. Thus, the hyperactivation of Th2 is similar to the response that is triggered during a parasitic (extracellular) infection or an allergic reaction, with the predominance of skin changes that are also found in schizophrenia [14,15,16]. Oxidative stress is used by our inflammatory cells to fight potential pathogens, and their values correlate with each other [14]. Therefore, there was an observed reduction of the anti-oxidant enzymes (catalase -CAT, superoxide dismutase -SOD, and peroxidase) that may be due to the transformation of the Th1 to Th2 immuno-response in schizophrenia patients [14]. It may be that further research is needed to observe Th2 immune response and its related oxidative changes as markers of schizophrenia, however not at the HPA axis immune peak response.

Brain-derived neurotrophic factor (BDNF) regulates the process of neurogenesis and synaptogenesis and can be carried across the blood–brain barrier. The serum BDNF levels are decreased in medicated and drug-naïve patients with schizophrenia [17,18]. Furthermore, BDNF levels are associated with cognitive impairment, particularly in chronic schizophrenia [19]. As studies on BDNF suggest its potential role in the pathogenesis and treatment of both neurological (e.g., Parkinson’s disease, Alzheimer’s disease, stroke) and psychiatric disorders (e.g., depression, schizophrenia) [20], its use as a schizophrenia marker seems to be controversial. Other peripheral psychosis markers, such as raised mRNA levels, breath test for ammonia and ethylene, have also been suggested [21] but seem to have low specificity. As none of the suggested tests alone were a real test for schizophrenia, a blood multitest for schizophrenia has been developed by means of 51 biomarkers (called VeriPsych) [22]. Unfortunately, it also had an unsatisfactory specificity. 

Peripheral gene expression may credibly be identified in blood, and schizophrenia is considered to be highly heritable. The Disrupted-in-Schizophrenia 1 (DISC1) gene regulates the neurodevelopmental processes and neural cell signalling [23]. Variations in several DISC1 polymorphisms have been associated with brain structure, changed brain function and impaired cognitive performance as well as with clinical severity of FEP [23]. Moreover, DISC1 polymorphism, as with many other genes, has been associated with not only risk of schizophrenia, but also with other mental disorders, such as schizoaffective disorder, bipolar disorder, major depression, autism and Asperger‘s syndrome [24]. A recent transcriptomic study of FEP revealed that a total of 978 genes were variously expressed and enhanced for possible pathways related to immune function and mitochondria in FEP [25]. 

The use of metabolomics is a modern approach towards the recognition of biomarkers of schizophrenia. A review of 63 studies investigating the metabolite biomarkers of schizophrenia revealed decreased levels of essential polysaturated fatty acids (EPUFAs), particularly arachidonic acid, vitamin E and creatine as well as increased levels of lipid peroxidation metabolites and glutamate [26]. Using a biomarker panel allows for the best possible distinction between patients and controls, and allows us to obtain a better understanding of the influence of antipsychotic treatment in the early stages of schizophrenia [27]. A recent biomolecule profile showed increased levels of asparagine, glutamine, methionine, ornithine and taurine, and reduced levels of aspartate, glutamate and alpha-aminoadipic acid (alpha-AAA) [27]. It has also been proposed to use blood-related protein biomarker methods in the study of psychiatric disorders due to the emerging evidence of solid molecular changes in psychiatric patients [28]. As studies generally confirm the role of lipid and amino acids rather in neuronal development and transmission [26,27,28,29,30] rather than as markers of schizophrenia, more studies are needed to check their applicability in diagnostics.

A meta-analysis of the cerebrospinal fluid (CSF) markers of the inflammatory state in schizophrenia and affective disorders found an increased CSF/serum albumin ratio and total CSF protein in schizophrenia and affective disorder [30]. In addition, in schizophrenia, the IgG level was increased, IgG to albumin ratio was decreased, and interleukin-6 (IL-6) levels and IL-8 levels were elevated [31]. These findings show that schizophrenia and affective disorders may present CSF abnormalities, involving blood-barrier disruption and inflammation, and further CSF studies should separate biochemical effects of affective from non-affective psychosis/schizophrenia.

Pillinger et al., in their systematic meta-review, summarized the effect sizes for the central nervous system (CNS: brain structural, neurophysiological as well as neurochemical parameters) and non-CNS (immune, cardiometabolic and hypothalamic-pituitary-adrenal (HPA) axis) dysfunctions in first-episode psychosis (FEP) [7]. Non-CNS abnormalities had a similar effect size as CNS abnormalities in FEP, and non-CNS abnormalities could be a cause or result of CNS dysfunction in psychotic states or their epiphenomena [7]. According to the author’s suggestion [7], it can be assumed that psychosis comprises multiple systems from illness onset, but it is not sufficient to name it as a multisystem disorder. However, non-CNS abnormalities can reflect CNS dysfunction, and should be involved in further marker studies.

### 3.2. Endophenotype

The endophenotype construct has been provided by studies that were based on neurophysiological, endocrinological, neuroanatomical, biochemical, cognitive/neuropsychological methods and these studies have pointed to a susceptibility to schizophrenia and that has been reproducible in patients’ families [32,33]. Therefore, the endophenotype is known to be partly hereditary and regulates the susceptibility to the disease [33]. To endophenotypes of schizophrenia have been included: sensory motor gating, oculomotor imbalance, impaired P300 event-related potential (ERP) and neurocognitive dysfunctions (working memory and/or information processing speed, executive functions and attention) [31,32,33,34]. Impaired eye movements and abnormalities of saccades have been associated with schizophrenia [33,34], and smooth pursuit eye movements have been proposed as biomarkers for the schizophrenia spectrum [35]. Saccadic abnormalities have also been displayed in clinical schizophrenia and pre-clinical high-risk groups, which provided proof for the consideration of saccadic abnormalities as possible neurobiological markers for schizophrenia [33]. Neurological soft signs (e.g., small motor discoordination, impairment in sequence-, balance-, and sensory- related integrations) [35] and impaired eye movements have also been found to be related with abnormalities in the cortex morphology [35,36,37,38] and could represent a genetically transmitted vulnerability factor to psychosis [36,39]. Although neurological soft signs, similar to impaired eye movements, have been widely described in schizophrenia patients and in full-siblings [33,39,40,41], some of these characteristic changes can be more specific for schizophrenia per se or for the schizophrenia spectrum. Therefore, it seems to be that memory guided saccades are rather specific to the schizophrenia spectrum, whereas antisaccades and neurological soft signs are rather the best predictors for schizophrenia disease [42].

The neurophysiological, neuropsychological and occulomotor disturbances, considered as an endophenotype, seem to be especially helpful tools in the diagnosis of schizophrenia when clinical symptoms are ambiguous.

### 3.3. Neuroimaging Biomarkers

Brain models in translational neuroimaging have concentrated on diagnosis and the identification of brain signatures that distinguish patients from healthy controls in order to establish a potential neurobiological index for the disorder [43]. Other neuroimaging models have been developed for risk assessment, early detection, predicting conversion to disorder, differential diagnosis, subtyping of patients and predicting the treatment response [43]. 

Andreou and Borgwardt have summarized the current research on neuroimaging biomarkers for predicting a transition in high-risk individuals [44]. Structural neuroimaging studies show that grey matter volume reductions have been described in high-risk patients compared with healthy controls in hippocampal/parahippocampal areas, cingulate cortex, medial and lateral frontal cortex and medial parietal cortex. Certain areas have been associated with later psychosis in high-risk persons (frontal cortex, anterior cingulate, temporal cortex, parietal cortex, cerebellum and insula). The high-risk subjects with later transition to psychosis show larger pituitary volumes than subjects without later psychosis. High-risk subjects with later transition to psychosis have also presented greater long-term volume reduction and cortical thinning in frontal areas, temporal areas, the parahippocampal and fusiform cortex, cingulate cortex, cerebellum, the medial and superior parietal lobes as well as the praecuneus and insula [44]. In addition, high risk subjects have displayed reduced whole white matter volume and several abnormalities in connectivity, including decreased fractional anisotropy of the inferior fronto-occipital fasciculus, and the inferior and superior longitudinal fasciculus [44]. Furthermore, functional MRI studies in high-risk subjects describe abnormalities in brain regional and functional connectivity during a variety of cognitive tests (working memory, verbal memory and fluency, social cognition, salience processing, and evidence gathering) [44]. 

Brain abnormalities can be observed in longitudinal magnetic resonance imaging studies at different stages of schizophrenia [45]. Studies from pre-clinical stages have reported more prominent cortical grey matter loss (superior temporal and inferior frontal regions) in persons who later had a transition to psychosis. Patients with FEP exhibited a decrease in multiple grey matter regions (frontal cortex and thalamus) over time as well as progressive cortical thinning in the superior and inferior frontal cortex. Patients with chronic schizophrenia process showed that grey matter decreased to a greater degree (frontal and temporal cortex, thalamus and cingulate cortex)—especially in poor-outcome patients [45]. Mouchlianitis et al. published a review of brain imaging studies of treatment-resistant schizophrenia with different modalities: structural MRI, fMRI, SPECT/PET, MRS, EEG [46]. Replicated differences in treatment resistant processes compared with responsive patients involved reductions in grey matter and the perfusion of frontotemporal regions, as well as increases in white matter and basal ganglia perfusion. Moreover, clozapine treatment has been shown to lead to a reduction in caudate nuclei volume [46]. 

In vivo neuroimaging studies allow the quantification of dopaminergic and glutamatergic function in the brain (38). Positron emission tomography (PET) and single photon emission computed tomography (SPECT) research studies have shown increased D2/D3 receptor density in schizophrenia, no evidence for a difference in striatal dopamine active transporter (DAT) receptor density in persons with schizophrenia and a large elevation in dopamine synthesis in schizophrenia compared with controls. The dopamine synthesis capacity is increased in individuals suffering from prodromal symptoms of schizophrenia and increases further with the onset of acute psychosis [47]. 

Glutamate and glutamine have been reported to exhibit regional abnormalities in high-risk subjects, with decreased concentrations in the thalamus and increased concentrations in the prefrontal cortex and the striatum—associated with later transition [44]. Proton magnetic spectroscopy (1H MRS) has been used to check glutamate and glutamine levels in vivo. A meta-analysis of proton magnetic resonance spectroscopy studies about schizophrenia found a significant rise in glutamate in the basal ganglia, glutamine in thalamus and Glx (glutamine and glutamate in combination) in the basal ganglia as well as in the medial temporal lobe. Elevated medial frontal Glx levels were evident in persons at high risk for schizophrenia, whereas elevated Glx in the medial temporal lobe was found in chronic schizophrenia [48]. Findings from another meta-analysis, show that higher glutamate levels in both medial frontal cortex and medial temporal lobe in patients with schizophrenia were associated with more severe symptoms and lower functioning, providing further support for the use of glutamatergic measures as a potential biomarker of illness severity [49]. Moreover, meta-analysis of brain chemistry in schizophrenia revealed decreased levels of N-acetyl aspartate (marker of neuronal integrity) levels in the frontal lobe, temporal lobe, and thalamus, in FEP and chronic schizophrenia [50]. 

Regarding neurophysiological changes, FEP is associated with decreased auditory P300 amplitude, and reduced duration–deviant mismatch negativity [7]. Meta-analysis of in vivo PET imaging studies of microglial activation in persons with schizophrenia, when compared with healthy controls, found moderate elevations in 18-kDa translocator protein (TSPO) binding in grey matter in schizophrenia when the binding potential (BP) was used as an outcome measure [51].

Generally, neuroimaging studies seem to point to nonspecific grey matter volume reductions. Observed increased D2/D3 receptor density in brain and glutamate/glutamine (Glx) changes in the frontal and temporal lobes, from which Glx is consistent with the neurotoxicity of psychosis, seem to be helpful in further diagnostics. 

### 3.4. Multimodal Approach

The Consensus Report of the APA Work Group on Neuroimaging Markers of Psychiatric Disorders from 2012 states that there are currently no identified brain imaging biomarkers that could be clinically useful for any diagnostic category in psychiatry [52]. The neuroimaging biomarkers currently used in clinical practice concern mainly their use in the diagnosis of neurocognitive disorders (i.e., dementia) [53]. Aydin et al., in their review, concluded that neuroimaging data that consider schizophrenia, bipolar disorder, and major depressive disorder, have been progressively gathered but are not yet mature enough for translation into clinical practice. The data for potential biomarkers for schizophrenia require further research, as there are inconsistencies and lack of replication studies, small sample size, insufficient longitudinal studies and the lack of multimodal studies [53].

Most neuroimaging studies have also indicated differences between groups of patients with psychosis, and today we need to facilitate outcome prediction based on data from an individual patient [54]. Machine learning is an example of a multivariate statistical approach to address this issue. The multiple research sites are involved in increasing the sample sizes. Using more than one modality of neuroimaging may be useful for improving the prediction of outcomes. Integrating neuroimaging data with non-imaging measures that have independently given altered outcomes in psychosis may also strengthen the predictive power. PSYSCAN is an example of an ongoing multicentre study to collect neuroimaging, clinical, cognitive and genetic data to identify a candidate index for the prediction of outcomes and to develop new software for data analysis (see www.psyscan.eu, accessed on 4 April 2022) [54].

Li et al. have established a new neuroimaging biomarker for schizophrenia diagnosis, prognosis and subtyping upon functional striatal abnormalities (FSA) [55]. The FSA score can be calculated from the subject’s resting-state fMRI brain scan using advanced machine learning techniques [https://www.szbiomarkers.net/fsa/ accessed on 4 April 2022]. FSA has been able to discriminate persons with schizophrenia from controls with an accuracy exceeding 80% (sensitivity −79.3% and specificity −81.5%). Inter-individual variation in baseline FSA scores was significantly correlated with the antipsychotic treatment response in two longitudinal cohorts. Moreover, striatal dysfunction was the most severe in schizophrenia subjects, milder in bipolar persons and indistinguishable in healthy persons with depression, obsessive–compulsive disorder and ADHD. The loci of striatal hyperactivity summarized the spatial distribution of dopaminergic activity and the expression of polygenic risk for schizophrenia [55].

The examination of individuals with ultra-high risk (UHR) is necessary for identifying potential biomarkers for the onset of schizophrenia. A voxel-wise whole-brain functional degree centrality (FDC) analysis is a graphical determination of the total functional connectivity between a voxel and the rest of the brain, and it has been used to check the hub regions of the brain network. Conjunction analysis has demonstrated, in comparison with healthy controls, that both UHR and schizophrenia patients had significantly increased FDC of the medial prefrontal cortex (MPFC) and decreased FDC of the right fusiform gyrus (FG) [56]. These altered FDCs were significantly correlated with disorganisation symptoms in both UHR and schizophrenia patients. These findings suggest that FDC within the MPFC and the right FG had potential to be used as candidate biomarkers related to a conversion to schizophrenia [56].

The inhibitory deficits in the motor cortex in schizophrenia patients have been found by using short-interval intracortical inhibition (SICI) by transcranial magnetic stimulation (TMS). The study by Du et al. [57] indicated that a higher resting-state left prefrontal motor cortex functional connectivity, attended by a higher fractional anisotropy of the left corona radiata, predicts fewer inhibitory deficits in schizophrenia. The inhibitory deficits in the motor cortex may partly be mediated by a descending prefrontal influence. SICI may function as a biomarker indexing the inhibitory impairment at the anatomic and circuitry points of schizophrenia [57].

Single-mode studies may detect single dimensional information and miss crucial differences between the patients and healthy subjects [58,59,60,61,62]. The multimodal explanation of functional MRI (fMRI), normal/structural MRI (sMRI) and diffusion tensor imaging (DTI) might provide a more powerful tool for the diagnostic process of schizophrenia [60]. The results of a study by Guo et al. [60] show that fMRI had the most significant feature. The fusion of these modalities also provided the most abundant information and the best predictive accuracy—86%.

The concept of biotypes may be another approach in determining the biomarkers of psychosis. This project was based on neurobiological heterogeneity in psychosis to identify subgroups independent of their symptomatology [63]. A substantial biomarker panel, including neuropsychological, stop-signal, saccadic/eye control as well as auditory stimulation paradigms, was acquired in persons suffering from schizophrenia, schizoaffective disorder and bipolar disorder with accompanying psychosis, their FDRs, and in healthy subjects. The multistep multivariate analyses identified three neurobiologically distinct psychosis biotypes: Biotype1—poor cognitive and sensorimotor functions; Biotype2—moderate cognitive dysfunction and sensorimotor hyper-reactivity; and Biotype3—almost normal cognitive and sensorimotor functions [63]. In addition, whole brain and regional grey matter density (GMD) biomarkers were studied as independent predictors of a biotype psychosis construct [64]. Biotype1 was characterised by the loss of an extensive and diffusely distributed GMD, with the largest changes in the frontal, anterior and middle cingulate cortex, as well as temporal regions; Biotype2—intermediate and more localised loss, with the largest changes in insula and frontotemporal areas; and Biotype3—small loss, localised to anterior limbic regions. GMD described specific brain structure characteristics in biotypes, corresponding to their cognitive and sensorimotor characteristics, and provided more powerful differentiation for biologically-related biotypes than symptom-related diagnosis [64].

The bioinformatic platform was developed in the EU project METSY (2013–2018) for the investigation of biological aetiologies in persons at risk of psychosis and in the FEP [65]. The METSY project was initiated with the purpose of identifying and evaluating multimodal peripheral and neuroimaging (bio)markers that could be able to predict the onset and prognosis of psychiatric illness and other pathological states e.g., metabolic symptoms.

The current progress of research on the diagnosis of schizophrenia indicates a necessary trend of using the multimodal approach in future marker studies, which could significantly increase the accuracy of diagnosis.

## 4. Conclusions

There is growing evidence of central and peripheral nervous system abnormalities in schizophrenia patients including individuals with ultra-high risk or antipsychotic naïve FEP. Upon the hypothesis of schizophrenic development, miscellaneous biomarkers were studied and described. Nevertheless, the evaluation of 168 studies that investigated biomarkers, by their statistical reliability and clinical effect-size, showed that only one of these passed the a priori threshold for clinical application [66]. The C allele of the 6672G>C single nucleotide polymorphism (SNP) in the HLA-DQB1 region predicted the risk of clozapine-related agranulocytosis with the OR 16.8 [67].

According to Kraepelin’s nosological principles, careful study of the clinical features is the most important point of diagnosis [68] but currently there is a need for this to be enriched with biomarkers. Of the described potential biomarkers, endophenotypes (neurophysiological, neuropsychological and occulomotor disturbances) seem to be helpful diagnostic tools. The neuroimaging glutamate/glutamine and D2/D3 receptor density changes, as well as immunological Th2 and PRL levels, as potential biomarkers, need further accuracy tests. The usefulness of potential markers from various body fluids should be verified with the appropriate time of body fluid collection, to minimize the influence of the stress axis on their biochemical concentrations. Instead of searching for a single biomarker of psychosis, researchers should focus on a multimodal approach.

## Data Availability

Not applicable.

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
