# Peer review of "Markers of Schizophrenia—A Critical Narrative Update"

_jcm, 2022, doi:10.3390/jcm11143964_

Round 1
Reviewer 1 Report
The authors reviewed biomarkers for schizophrenia based on biochemical, neurophysiological, neuropsychological, neuroimaging, and multimodal findings. To exclude the long-term impacts of treatment and chronic illness, the included evidence was mainly from research on individuals with ultra-high risk or antipsychotic naïve first-episode patients.
Although various types of biomarkers for schizophrenia have been reviewed/summarized by several recent papers, e.g., ref 9, 13, 34, 35, the authors here provided a concise and comprehensive review of markers for schizophrenia in a broad range. This is a timely review and a critical update which may benefit the researchers in this field with an overall picture of the markers for schizophrenia. I have only a few minor comments as follows:
1, in both the abstract and conclusion paragraph, the authors addressed the role of body fluid collection in affecting biochemical/immunological markers. However, this aspect was implied in the Results and Discussion but not explicitly discussed. If the authors consider this is critical, maybe better to discuss this in a separate paragraph/subsection?
2, Sub-sections in the ‘3. Results and Discussion’ were all numbered as 3.1. For Endophenotype, Neuroimaging biomarkers, and Multimodal approach, they may be numbered as 3.2, 3.3, and 3.4, respectively.
Author Response
Although various types of biomarkers for schizophrenia have been reviewed/summarized by several recent papers, e.g., ref 9, 13, 34, 35, the authors here provided a concise and comprehensive review of markers for schizophrenia in a broad range. This is a timely review and a critical update which may benefit the researchers in this field with an overall picture of the markers for schizophrenia. I have only a few minor comments as follows:
1. in both the abstract and conclusion paragraph, the authors addressed the role of body fluid collection in affecting biochemical/immunological markers. However, this aspect was implied in the Results and Discussion but not explicitly discussed. If the authors consider this is critical, maybe better to discuss this in a separate paragraph/subsection?
-Now we discussed these issues in lines 58-63, 70-76, 85-91, 103-106, 116-118, 151-154, 180-182, 254-257, 344-346.
2. Sub-sections in the ‘3. Results and Discussion’ were all numbered as 3.1. For Endophenotype, Neuroimaging biomarkers, and Multimodal approach, they may be numbered as 3.2, 3.3, and 3.4, respectively.
-Done.
Reviewer 2 Report
abstract:
line 8: first statement: I suggest rewriting the first sentence in a more gentle way... eg. "associated with functional impairment"-- introduction line 25: please provide a reference line 27: please provide a reference (In general, each statement would require a reference) line 29 please change to "polygenic" lines 38-41 I would suggest not writing the same sentence already present in the abstract -- Methods Authors should declare should state whether the review is narrative or systematic (in this case providing inclusion and exclusion criteria) --- Results: the results/discussion section contains a list of results, but these are not critically discussed. Eg. the authors declare that prolactin levels are elevated in antipsychotic naïve patients with schizophrenia but the relevance of hyperprolactinaemia as a marker is not discussed here or elsewhere. the same applies to other biomarkers line 74: SOD has been repeated --- Conclusion "According to Kraepelin’s nosological principles, careful study of the clinical features is more important than neuropathological and etiological studies". I find this conclusion may be captious for several reasons; the clinical phenotype is subject to high heterogeneity even in patients with Mendelian monogenic disorders, sharing the same genetic and biological basis, let alone in polygenic disorders with variable penetrance, thus the study of clinical features may not provide sufficient information for tailored treatment; conclusions are similar to what has been stated in the introduction, not adding novelty to the field and not indicating future directions for scientific research
Author Response
Comments and Suggestions for Authors
--abstract:
line 8: first statement: I suggest rewriting the first sentence in a more gentle way... eg. "associated with functional impairment"
-corrected
-- introduction
line 25: please provide a reference
-A reference by Owen at el (2011) now is added.
line 27: please provide a reference (In general, each statement would require a reference)
- A reference by Maj at el (2021) now is added.
line 29 please change to "polygenic"
-corrected.
lines 38-41 I would suggest not writing the same sentence already present in the abstract
-Corrected.
-- Methods
Authors should declare should state whether the review is narrative or systematic (in this case providing inclusion and exclusion criteria)
-The review is narrative and this information now is added to the title, introduction and methods.
--- Results:
the results/discussion section contains a list of results, but these are not critically discussed. Eg. the authors declare that prolactin levels are elevated in antipsychotic naïve patients with schizophrenia but the relevance of hyperprolactinaemia as a marker is not discussed here or elsewhere. the same applies to other biomarkers
-Now we discussed these issues in lines 58-63, 70-76, 85-91, 103-106, 116-118, 151-154, 180-182, 254-257, 344-346.
line 74: SOD has been repeated
-deleted.
--- Conclusion
"According to Kraepelin’s nosological principles, careful study of the clinical features is more important than neuropathological and etiological studies". I find this conclusion may be captious for several reasons; the clinical phenotype is subject to high heterogeneity even in patients with Mendelian monogenic disorders, sharing the same genetic and biological basis, let alone in polygenic disorders with variable penetrance, thus the study of clinical features may not provide sufficient information for tailored treatment;
-We rephrased Conclusion subsection according to suggestion.
conclusions are similar to what has been stated in the introduction, not adding novelty to the field and not indicating future directions for scientific research
-Now, the rephrased conclusion contains new informations that address new directions added to the field.
Round 2
Reviewer 2 Report
I am very satisfied with the corrections made. I recommend that authors check their drafts carefully to correct typos in the proof-checking phase (e.g. line 257 'neurotoxicity').